# Effects of microsurface structure of bioactive nanoparticles on dentinal tubules as a dentin desensitizer

**Yang-Jung Choi[1], Moon-Kyoung Bae[2], Yong-Il Kim[3], Jeong-Kil Park[1], Sung-Ae Son[1] \***

**1** Department of Conservative Dentistry, School of Dentistry, Pusan National University, Dental Research Institute, Yangsan, Gyeongsangnam-do, Korea, **2** Department of Oral Physiology, School of Dentistry, Pusan National University, Yangsan, Gyeongsangnam-do, Korea, **3** Department of Orthodontics, Dental Research Institute, Pusan National University Dental Hospital, Yangsan, Gyeongsangnam-do, Korea

\* songae76@gmail.com

## Abstract

In this in vitro study, spherical mesoporous bioactive glass nanoparticle (MBGN) and non-porous bioactive glass nanoparticle (BGN) were fabricated. The impact of mesopores on dentinal tubule occlusion and bioactivity was compared to examine the potential of these materials in alleviating dentine hypersensitivity (DH). MBGN, dense BGN were synthesized by sol-gel methods and characterized. Bioactivity and ion dissolution ability were analyzed. Twenty-four simulated sensitive dentin discs were prepared and randomly divided into three groups (n = 8 each); Group 1, no treatment; Group 2, Dense BGN; Group 3, MBGN. Then, four discs per group were treated with 6wt.% citric acid challenge to determine the acidic resistance. The effects on dentinal tubule occlusion were observed by FESEM. The micro-tensile bond strength (MTBS) was also measured. Cytotoxicity was examined using the MTT assay. According to the results, dense BGN without mesopore and MBGN with meso-pore were successfully fabricated. Dense BGN and MBGN occluded the dentinal tubule before and after acid challenge. However, only MBGN formed a membrane-like layer and showed hydroxyapatite formation after soaking SBF solution. There were no significant differences in MTBS among dense BGN, MBGN (P>0.05). The cell viability was above 72% of both materials. The higher bioactivity of MBGN compared with that of dense BGN arises from the structural difference and it is anticipated to facilitate dentin remineralization by inducing hydroxyapatite deposition within the dentinal tubule.

## 1. Introduction

Dentin hypersensitivity (DH) generally occurs upon external stimulation from steam, heat, and contact with the dentinal tubule that is exposed as a result of gingival recession or enamel loss caused by abrasion, attrition, or acid erosion [1]. The incidence of DH ranges from 10–30% and commonly affects women in their 20s–50s. DH frequently occurs in canines and first premolars and is characterized by momentary and sharp pain upon external stimulation [2]. Among various theories pertaining to the etiology of DH, the most widely accepted theory is

**Data Availability Statement:** All relevant data are within the paper and its Supporting Information files.

**Funding:** unfunded study.

**Competing interests:** No authors have competing interests.

the hydrodynamic theory. Proposed by Brannstrom and Astrom, the hydrodynamic theory posits that thermal, osmotic, and physical stimuli induce movement of fluid within the dentinal tubule and stimulate the nerve endings. Such nerve ending activation is known to cause sharp, rapid pain, and many treatments primarily aim to occlude the dentinal tubule to prevent this activation [3, 4].

Products that alleviate DH include calcium-containing gum, toothpaste, mouthwash, and professional fluoride varnish, but their effects are transient, and the products need to be repeatedly applied [5]. Recently, a remineralization solution containing potassium oxalate and ferric oxalate was introduced, and it has been confirmed to mitigate DH by occluding exposed dentinal tubules. However, the limitation of the short duration of the occluding effect remains [6–9]. Furthermore, dentinal tubules were found to be re-exposed upon exposure to dietary acid [10, 11]. Therefore, biomaterials used to treat DH must have practical occluding effects for dentinal tubules and have long-term stability even after exposure to dietary acid [12, 13]. In general, DH is accompanied by hard tissue defects. Such cases require additional resin restoration, and the desensitizing treatment must not have a negative impact on the bond strength between the restoration material and teeth [14].

Bioactive glass nanoparticles (BGN) show large surface area, high intensity of illumination, hydrophilicity, and wettability while having cell adhesive capacities and high bioactivity in host tissue [15, 16]. Once BGN powder comes into contact with a biological fluid, such as saliva, it releases sodium and calcium ions and increases pH, thereby creating an alkaline environment [17]. In an alkaline environment, calcium and phosphate ions form an apatite that occludes the dentinal tubule and diminishes fluid movement within the dentinal tubule [18]. Recent studies reported successful dentinal tubule occlusion and dentin remineralization using BGN. Besinis et al. [19] applied bioactive hydroxyapatite (HA) and silica nanoparticles on dental specimens and confirmed that they infiltrate the dentinal tubule and cause remineralization in the dental specimens.

Mesoporous bioactive glass (MBG) is a bioactive glass (BG), with 2–50 nm mesopores, that has a larger surface area compared to non-porous BG. Owing to such features, it has a relatively higher response rate, dissolution capacity, and property to induce re-precipitation of apatite materials [20, 21]. Moreover, a recent study reported that MBG has better biocompatibility than BG [17]. In a comparison of the bioactivities of MBG and BG scaffolds, Zhu et al. [22] reported that calcium and phosphate ions aggregate and form nucleation on the surface of MBG better than on BG and MBG is, therefore, associated with quicker crystallization and better bone-forming bioactivity. Considering the size of dentinal tubules near enamels, applying nanosized particles to dentinal tubules appears to have increased the dentinal tubule-occluding effects. In 2019, Yuan et al. [23] confirmed that applying nanosized HA particles to dentinal tubules increases the proportion of dentinal tubule occlusion. However, studies that compare the dentinal tubule-occluding capacity and bioactivity of mesoporous bioactive glass nanoparticles (MBGN) and non-porous BGN are rare. In this context, this study aims to synthesize spherical MBGN and non-porous BGN and compare the impact of mesopores on dentinal tubule occlusion and bioactivity to examine the potential of these materials in alleviating DH.

## 2. Materials and methods

Dense BGN and MBGN was synthesized by sol-gel method and characterized by X-ray diffraction (XRD), Fourier transform infrared (FT-IR) spectroscopy, $N_2$ adsorption-desorption isotherms, field-emission scanning electron microscopy (FESEM). In vitro bioactivity, ion dissolution ability, dentinal tubule occlusion ability, effect on microtensile strength of composite resin material and cytotoxicity of both bioactive glasses were evaluated (Fig 1).

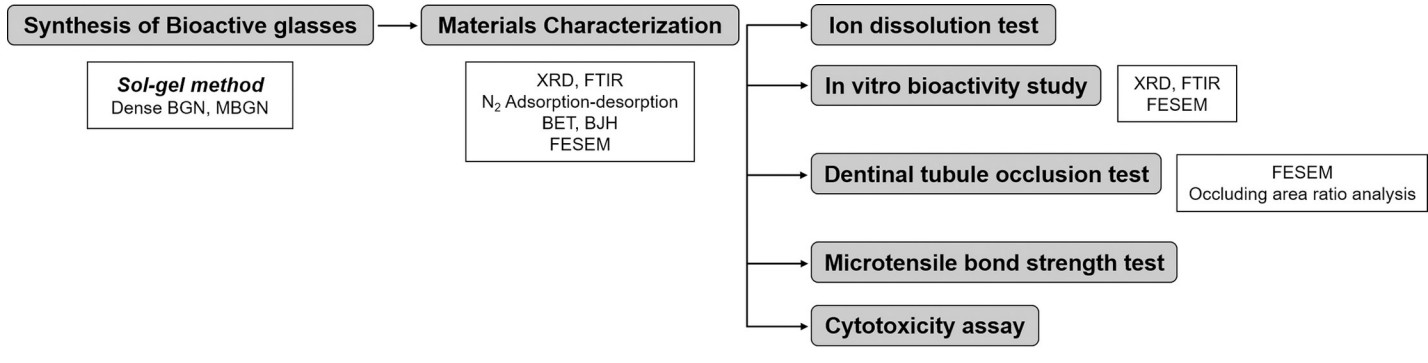

**Fig 1. A schematic diagram of materials and method.**

## 2.1 Synthesis of bioactive glass

**2.1.1. Synthesis of dense bioactive glass nanoparticles and mesoporous bioactive glass nanoparticles.** Dense BGN was synthesized using the sol-gel method (Fig 2A). A solution containing 5 ml tetraethyl orthosilicate (TEOS) (Sigma-Aldrich, St. Louis, MO, USA) and 0.25 ml triethyl phosphate (TEP) (Sigma-Aldrich, St. Louis, MO, USA) in 50 ml ethanol (Samchun, Pyeongtaek, South Korea) was added to a solution consisting of 16.24 ml ethanol and ammonium hydroxide solution (Samchun, Pyeongtaek, South Korea) and was continuously stirred for 35 minutes. After adding 3.12 g ground calcium nitrate tetrahydrate (Sigma-Aldrich, St. Louis, MO, USA) to this solution, it was vigorously stirred for 85 minutes. The solution was then centrifuged at 7,350 rpm for 5 minutes, washed with ethanol, dried at 60˚C for 6 hours, and calcinated in a 600˚C furnace for 5 hours.

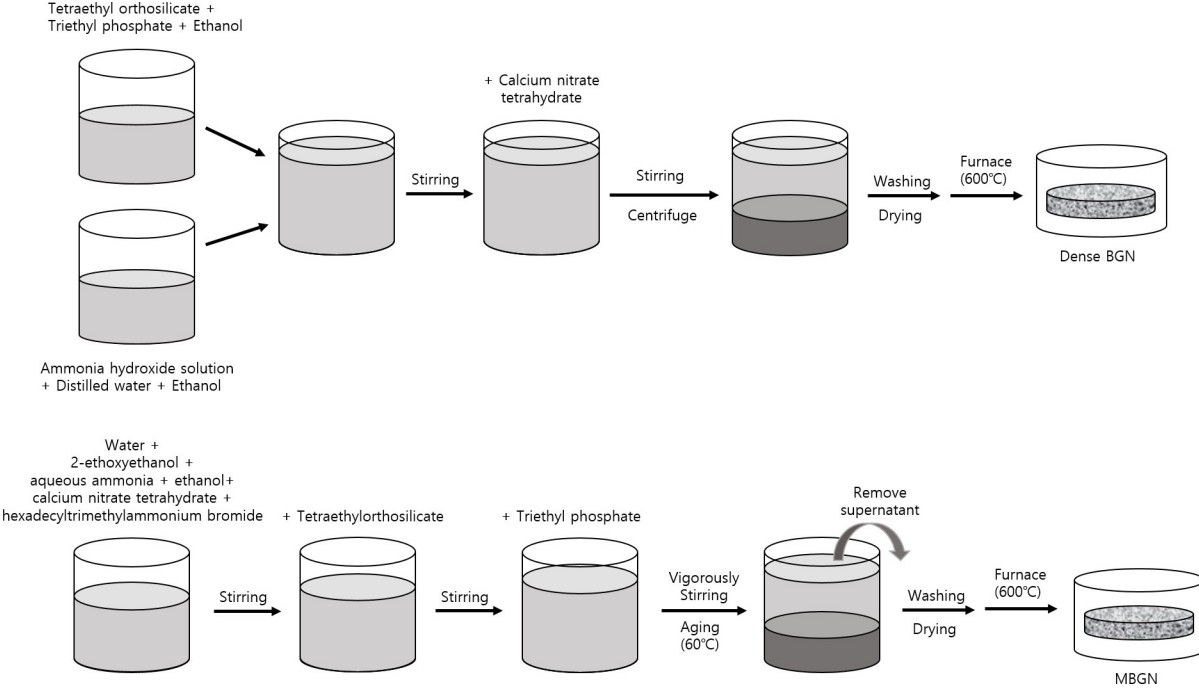

**Fig 2. An experimental visual form of bioactive glass nanoparticle by sol-gel method.** (a) Dense BGN, (b) MBGN.

MBGN was synthesized using the sol-gel synthesis protocol (Fig 2B). In a solution containing 150 ml water and 20 ml ethanol, 10 ml 2-ethoxyethanol (Sigma-Aldrich, St. Louis, MO, USA) was added as a co-solvent, and 1 g surfactant hexadecyltrimethylammonium bromide (CTAB) (Sigma-Aldrich, St. Louis, MO, USA), 2 ml catalyst aqueous ammonia (Samchun, Pyeongtaek, South Korea), and 3.12 g precursor calcium nitrate tetrahydrate were added and stirred for 30 minutes at room temperature. Subsequently, 5 ml TEOS was added to this solution and stirred for 30 minutes. After adding 0.25 ml triethyl phosphate to achieve a silicon dioxide to phosphate ratio of 60:4, the solution was vigorously stirred for 4 hours to obtain white precipitation and dried at 60˚C for 24–48 hours. The precipitate was then calcinated in a 600˚C furnace for 5 hours.

## 2.2. Materials characterizations

The synthesized dense BGN and MBGN were characterized as follows. The crystalline phase was determined via x-ray diffraction (XRD) analysis using 40 kV, 40 mA Cu-Kα radiation, and an X-ray diffractometer (Ultima IV multipurpose XRD system, Rigaku, The Woodland, TX, USA). XRD analysis was performed with a scanning speed of 4˚/min and 2θ range of 10–70˚ for wide-angle XRD patterns, and with a scanning speed of 1.2˚/min and range of 0.5–10˚ for small-angle XRD patterns. Functional groups and chemical composition were examined via Fourier-transform infrared spectroscopy (FTIR) analyses using a Nicolet 5700 spectrometer (Thermo Scientific Inc., Madison, WI, USA). Information about $N_2$ adsorption-desorption isotherms was obtained at 77 K using an ASAP 2420 gas adsirotuib analyzer (Micromeritics, Atlanta, GA, USA). Specific surface area and pore size distribution were calculated via the BET and BJH method, respectively. The topography of the sample was observed and analyzed with field emission scanning electron microscopy (FESEM) (Sigma, Zeiss, Germany).

## 2.3. Ion dissolution test and in vitro bioactivity

Dense BGN and MBGN were placed in multiple channel pelleting molds, and 5 MPa pressure was applied for 60 seconds to produce 12 tablets of 10 mm diameter and 1 mm thickness. The mineralization experiment proceeded in compliance with the International standards (ISO/FDIS 23317). Stimulated body fluid 5X (SBF) (Biosesang, Seongnam, Korea) was prepared, and the SBF volume was calculated using the following formula:

$$V_s = S_a/10$$

where $V_s$ is the volume of SBF solution (mL), and $S_a$ is the surface of tablets ($mm^2$).

After placing 19 mL SBF in the tube, tablets were inserted and stored at 36.5˚C for 1, 3, 7, 14, and 42 days. Each tablet was then removed from the SBF solution, washed once with ethanol and twice with pure water to stop the mineralization process, and dried in an oven at 60˚C. The mineral composition and crystal structure of the hydroxyapatite formed on the surface of the tablets were analyzed with XRD (Rigaku, The woodland, TX, USA) and Fourier-transform infrared spectroscopy (PerkinElmer Inc., Waltham, MA, USA). Hydroxyapatite formed on the surface of the tablets was observed with scanning electron microscopy (SEM). ICP-OES (Optima 8300, Perkin Elmer, Waltham, MA, USA) was performed on the SBF solution used to soak the tablets to analyze the dissolved ions.

## 2.4. Dentinal tubule occlusion test

**2.4.1. Sample preparations and experimental design.** Twenty-four premolars were collected in a manner that was verified and reviewed by the Institutional Review of Pusan National University Dental Hospital (PNUDH-2019-040). The teeth were stored in 4˚C, 0.5%

thymol solution, and were used within 3 months of extraction. A low-speed diamond saw (Struers Accutom-50, Ballerup, Denmark) was used to section each tooth from the inferior of the enamel-dentinal junction vertically to the long axis under water cooling to obtain 1 mm tooth discs. The discs were polished for 60 seconds using 320- and 600-grit silicon carbide (SiC) polishing paper. After soaking the discs in 1 wt.% citric acid solution for 20 seconds to open the dentinal tubules for the fabrication of sensitive tooth models, they were rinsed with water spray. The specimens were randomized into one of the following 3 groups (n = 8).

Group 1: No treatment (control).

Group 2: Slurry prepared with a ratio of 100 mg Dense BGN/200 μL deionized water is placed in a rubber cup installed on a NiTi engine (X Smart Plus, Dentsply Sirona, Ballaigues, Switzerland), run for 15 seconds at 800 rpm and again for a further 15 seconds, a total of 30 seconds.

Group 3: Slurry prepared with a ratio of 100 mg MBGN/200 μL deionized water is placed in a rubber cup installed on a NiTi engine, run for 15 seconds at 800 rpm, and again for a further 15 seconds, a total of 30 seconds.

To determine the resistance to a strong acidic environment, four discs from each group were randomly selected and post-treatment was applied to the remaining discs. Each disc was soaked in 6 wt.% citric acid solution (pH 1.5) for 60 seconds and wholly rinsed with deionized water.

**2.4.2. FESEM evaluation of dentinal tubule occlusion.** After fabricating tooth discs using the method previously described, treated samples were prepared, and the surfaces were observed with FESEM. After washing each disc with a water spray and drying, the discs were sputter-coated using Au-Pd alloy. Two areas of each sample were observed with FESEM at x2,000, x5,000, and x10,000 magnification to obtain micrographs. Image J (NIH, Frederick, MD, USA) was used to calculate the occluded dentinal tubule ratio (the area of occluded tubules/total tubules area). To compare the occluded dentinal tubule depths by group, a groove was formed on the additionally prepared tooth samples and then longitudinally sectioned to expose the longitudinal surface. After performing sputter-coating via the same method described previously, the samples were observed via FESEM at x10,000 magnification.

## 2.5. Microtensile bond strength (MTBS) test

To assess the impact of each desensitizing treatment on immediate MTBS, an additional 15 teeth were collected, and the mid-coronal surfaces of the teeth were exposed by sectioning them vertically to the long axis under water cooling using a low-speed diamond saw (Struers Accutom-50, Ballerup, Denmark). The exposed surface was polished with 600-grit SiC paper and soaked in 1 wt.% citric acid solution to reproduce the sensitive tooth model. The samples were randomized into one of 3 groups (n = 5) and treated accordingly as described in 2.4.1, washed with water spray for 30 seconds, and dried. Clearfil SE bond (Kuraray Co., Osaka, Japan) was applied to the surface of each sample as per the manufacturer's instructions, and resin composite (Z250, 3M ESPE, St Paul, MN, USA) was applied in layers of 2 mm for a total thickness of 4 mm, followed by 20 seconds of photopolymerization using an LED curing unit (DEMI; > 1000mW/cm2; Kerr Corporation, Middleton, WI, USA). After storing them in deionized water at 37˚C for 24 hours, the bonded teeth were sectioned to obtain 4 beam specimens, 0.9 mm x 0.9 mm, for each tooth, a total of 20 specimens. Each beam was fixed on an MTBS tester (Bisco, Schaumburg, IL, USA) using a cyanoacrylate adhesive and tensile force was applied at 1 mm/min crosshead speed until failure occurred.

## 2.6. Cytotoxicity assay

Human dental pulp tissues were obtained from premolars extracted for orthodontic purposes, and human dental pulp cells (HDPCs) were extracted. Separated pulp tissues were cultured in a -modified essential medium (Dulbecco's modified Eagle's medium; LM001-01; Welgene, Seoul, South Korea). After adding 10% fetal bovine serum (Gibco, CA, USA), 100 units/mL penicillin, and 100 mg/mL streptomycin to the medium, pulp tissues were incubated in a humidified atmosphere at 37˚C and 5% $CO_2$. The medium was replaced every two days, and third passage cells were used. In vitro cytotoxicity of dense BGN and MBGN was examined with 3-[4,5-dimethylthiazol-2-yl]-2,5-diphenyltetrazolium bromide (MTT) assay. After applying dense BGN and MBGN (0, 10, 20, 40, 80, 160, 320 µg/mL distilled water (DW)) on a 48-well plate, the DW was evaporated. HDPCs were seeded at 10,000 cells/well and incubated at 37˚C and 5% $CO_2$ for 72 h. Subsequently, 5 mg/ml MTT (Sigma-Aldrich, St. Louis, MO, USA) was applied and incubated for 3 hours, followed by 5 minutes of washing with dimethyl sulfoxide (DMSO) (Sigma-Aldrich, St. Louis, MO, USA). The optical density was measured using the ELISA reader (Dynex, Chantilly, USA). This experiment was conducted in triplicate, and the results were presented as relative cell viability (%) against the control group for which only the culture medium was measured.

# 3. Results

## 3.1. Materials characterizations

**3.1.1. XRD and FTIR.** Fig 3 shows the XRD patterns of MBGN and dense BGN. In the wide-angle XRD patterns, neither MBGN nor dense BGN showed diffraction peaks other than the $SiO_2$ peak near $2\theta \approx 25˚$. In the small-angle XRD patterns, MBGN showed a peak near 2.5˚. In contrast, dense BGN did not show diffraction peaks in the small-angle XRD patterns.

Fig 4 shows the FTIR spectra of MBGN and dense BGN. Both MBGN and dense BGN show a silicate adsorption band in an asymmetric stretching mode near 1060, 800, and 470 cm⁻¹.

**3.1.2. Surface area and pore size análisis.** The porosity of MBGN and dense BGN was evaluated using $N_2$ adsorption-desorption isotherms. MBGN showed a type IV isotherm, while dense BGN showed a type II isotherm. In Fig 5, MBGN shows type H1 hysteresis loops. This suggests that pores of relatively equal size are evenly distributed. Table 1 shows the results of SBET (m²/g), Vp (cm³/g), and Dp (Å) of MBGN and dense BGN analyzed via the BET and

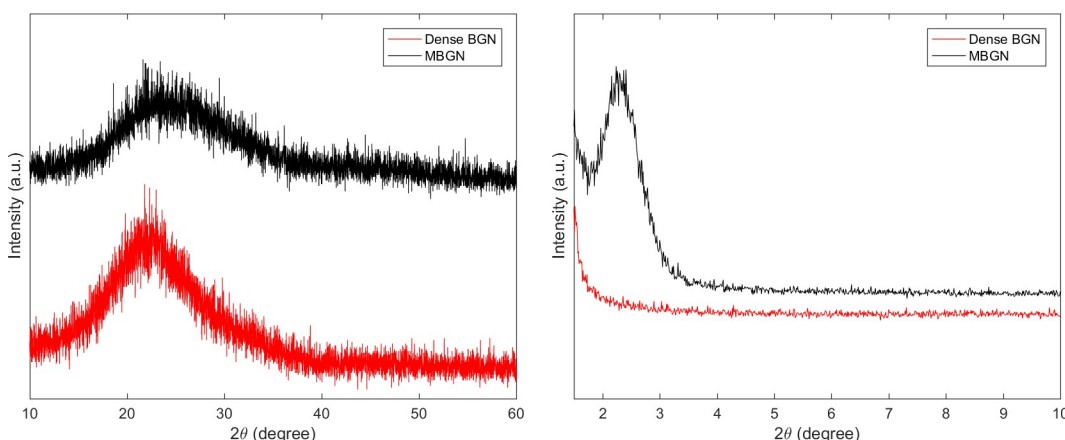

**Fig 3.** (a) Wide-angle XRD pattern of MBGN and Dense BGN, (b) Small-angle XRD pattern of MBGN and Dense BGN.

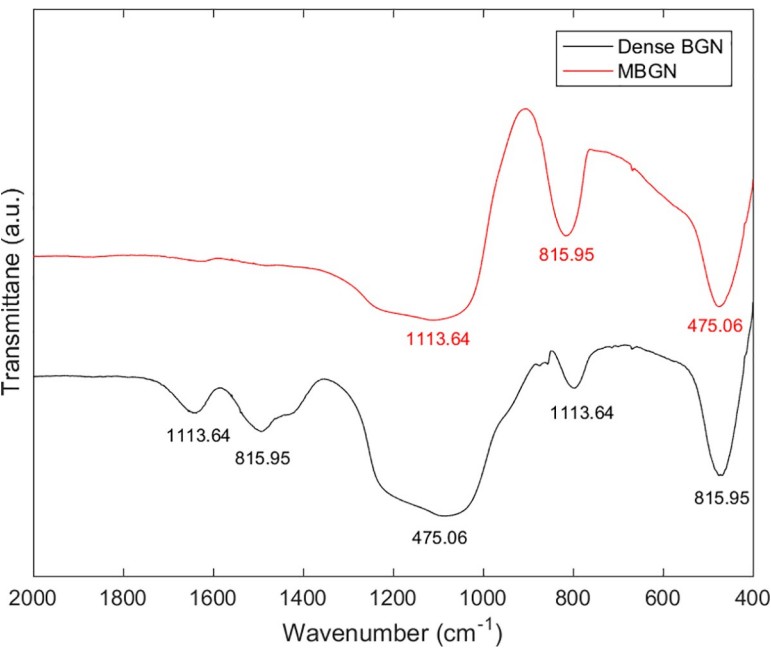

**Fig 4. FT-IR pattern of MBGN and Dense BGN.**

BJH methods. MBGN had a markedly greater surface area (152.40 m$^2$/g) than that of dense BGN (7.37 m$^2$/g).

**3.1.3. FESEM images.** The topology of the materials was analyzed using FESEM. On the FESEM image, MBN showed a size of 512–604 nm (Fig 6(A)), while dense BGN showed a size of 493–502 nm (Fig 6(B)). Whereas the surface of the BGN is irregular, that of the dense BGN is smooth.

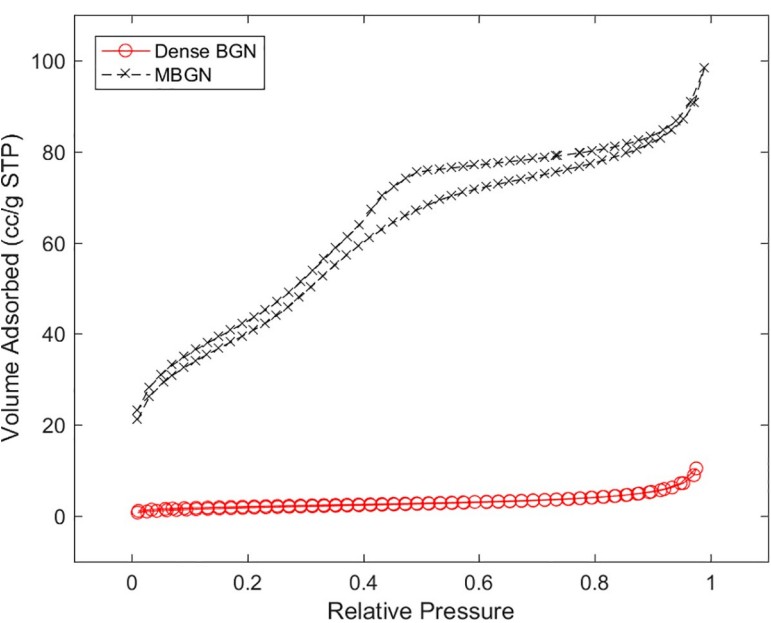

**Fig 5. Nitrogen adsorption-desorption isotherms of MBGN and dense BGN.**

**Table 1. N$_2$ adsorption results.**

| Samples | S$_{BET}$(m$^2$/g) | V$_P$(cm$^3$/g) | D$_P$(nm) |
|---|---|---|---|
| MBGN | 152.40 | 0.15 | 3.99 |
| Dense BGN | 7.37 | 0.02 | 13.11 |

S$_{BET}$(m$^2$/g), surface area; V$_P$(cm$^3$/g), total pore volume for pores with radius; D$_P$(nm), average pore radius.

## 3.2. Ion dissolution test

Ion dissolution of dense BGN and MBGN is shown in Fig 7. Calcium ion concentration decreased over time for both dense BGN and MBGN, with a more significant reduction in MBGN. In MBGN, calcium ion concentration decreased from 0.617 ppm to 0.078 ppm on 14 and 42 days, respectively, showing that it decreased to almost 0 ppm. Phosphate ion concentration increased over time. Dense BGN had a higher phosphate ion concentration than MBGN until 14 days, however after 42 days MBGN had a higher phosphate concentration. Silicon ion concentration gradually increased over time, with a more significant increase in dense BGN than in MBGN.

## 3.3. In vitro bioactivity study

**3.3.1. XRD analysis and FTIR.** Fig 8 shows a graph of the XRD patterns of MBGN and dense BGN soaked in SBF. In the pattern with the 2θ range set to 10˚–60˚, a SiO$_2$ peak (amorphous phase) was observed. MBGN samples soaked in SBF solution for 5, 14, and 42 days showed a peak for the hydroxyapatite phase (HAp, Ca$_{10}$ (PO$_4$)$_6$ (OH)$_2$, JCPDS #09–0432), while dense BGN samples did not show this peak.

Fig 9 shows a graph of FTIR patterns of MBGN and dense BGN samples soaked in SBF. The FTIR pattern of MBGN shows Si-O-Si asymmetric stretching and rocking vibration at 800 cm$^{-1}$. Si-OH bonding-related vibration is observed at 1644, 957 cm$^{-1}$, and CO$_3$$^{2-}$ related C-O peak (carbonated hydroxyapatite) is observed at 1428, 873 cm$^{-1}$. In the FTIR patterns of dense BGN, Si-O-Si asymmetric stretching and rocking vibration at 800 cm$^{-1}$ and Si-OH bonding-related vibration at 957 cm$^{-1}$ was observed.

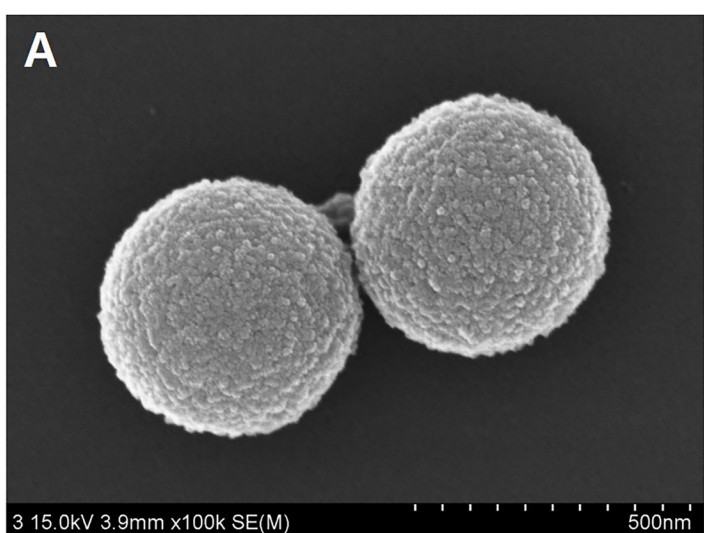
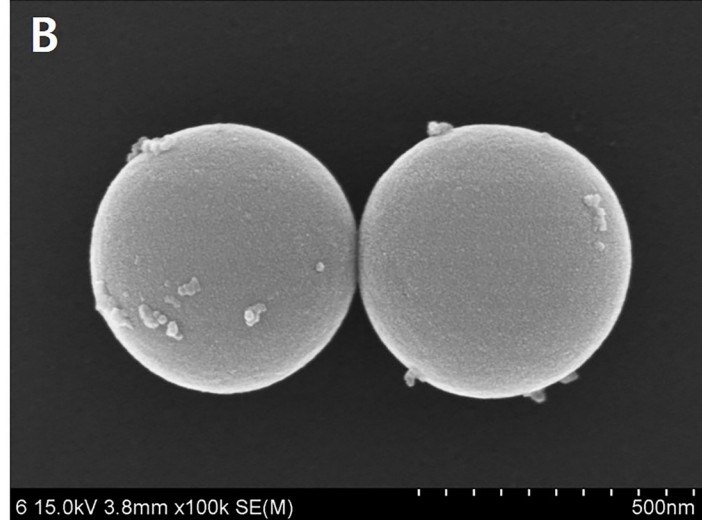

**Fig 6.** FESEM images of (a) MBGN and (b) Dense BGN.

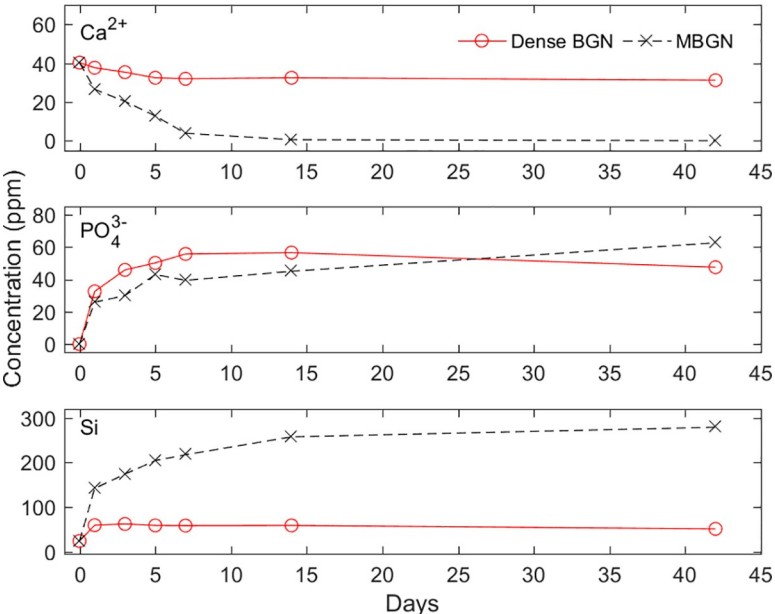

**Fig 7. Change in calcium, phosphate, silicon concentration over time.**

**3.3.2. FESEM images.**  Fig 10 shows the results of FESEM images of tablet forms of MBGN and dense BGN precipitated for 1, 3, 5, 7, 14, and 42 days in the SBF solution. With MBGN samples, the FESEM image of the sample precipitated in the SBF solution for 42 days shows hydroxyapatite in a flake-like form. With dense BGN samples, hydroxyapatite formation was not observed in any of the images of the samples.

## 3.4. Dentinal tubule occlusion test

**3.4.1. FESEM observation of tubule occlusion.**  Fig 11 shows the FESEM images of the surfaces and longitudinal sections of the disc teeth samples simulating hypersensitive teeth. Group 1 (Fig 11A1–11D1 ) has a dentin surface with no smear layer as the dentinal tubule was

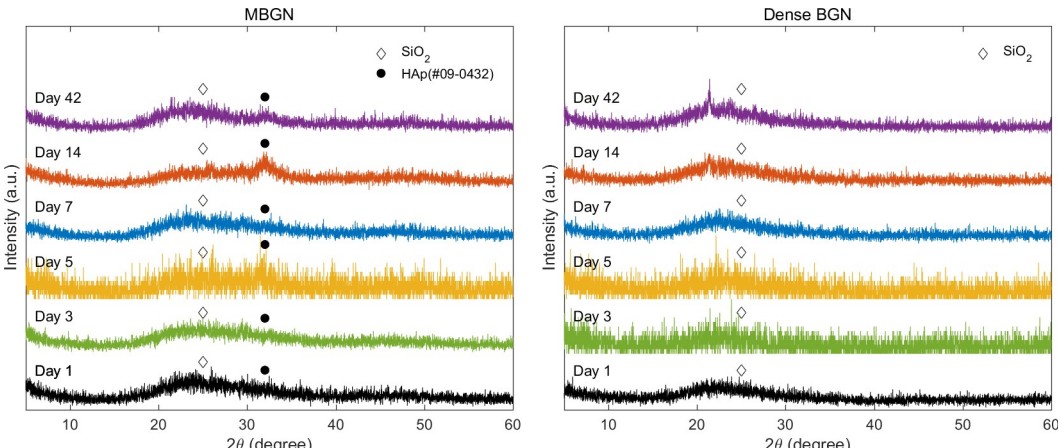

**Fig 8.**  X-ray diffraction patterns of (a) MBGN and (b) Dense BGN at 1, 3, 5, 7, 14, 42 days after soaking in simulated body fluid (SBF) solution.

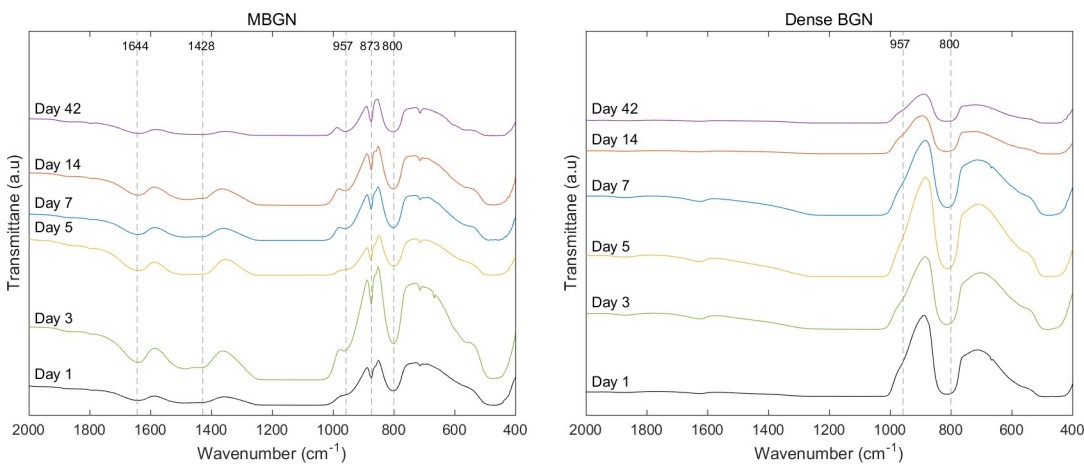

**Fig 9.** FT-IR patterns of (a) MBGN and (b) Dense BGN at 1, 3, 5, 7, 14, 42 days after soaking in simulated body fluid (SBF) solution.

## (a) MBGN

### 1days          5days          42days

## (b) Dense BGN

### 1days          5days          42days

**Fig 10.** FESEM images of (a) MBGN and (b) Dense BGN tablet at 1, 5, 42 days after soaking in simulated body fluid (SBF) solution.

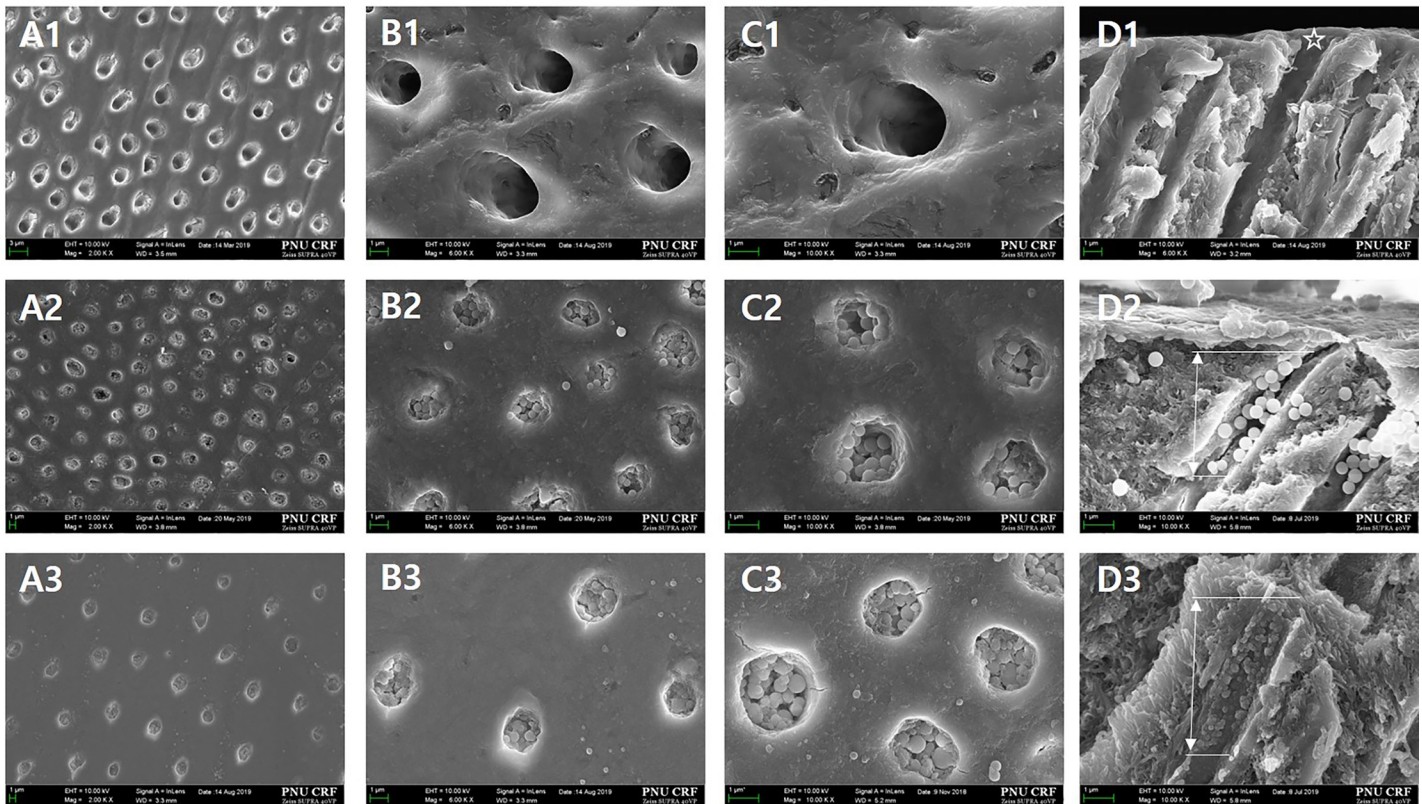

**Fig 11. FESEM micrographs of the top and longitudinal sections of the hypersensitive tooth model to confirm the tubule-occluding effect in each group (A, x2,000; B, x6,000; C, x10,000; D, sagittal).** There is no smear layer in the dentinal tubule in the A1–D1 images of Group 1 (control) (\*). The image of Group 2 (Dense BGN) shows a partially occluded dentinal tubule at 4–5 μm depth (arrow). The image of Group 3 (MBGN) shows a partially occluded dentinal tubule at 5–6 μm depth (arrow).

opened after soaking in 1 wt.% citric acid solution for 20 seconds. Group 2 (Fig 11A2–11D2) and Group 3 (Fig 11A3–11D3) are groups that were treated with dense BGN and MBGN, respectively, and the images show that the particles partially occluded the dentinal tubules.

FESEM observation of the surfaces and longitudinal sections of the tooth samples 14 days after exposure to 6 wt.% citric acid solution showed that the concentration of particles remaining on the dentin surface differed among the groups. Group 1 (Fig 12A1–12D1) had a larger diameter dentinal tubule compared to when the samples were treated only with 1 wt.% citric acid. Group 2 (Fig 12A2–12D2) and Group 3 (Fig 12A3–12D3) had a decrease in the concentration and depth of ions blocking the tubule. Compared to those in Group 2, the particles in the tubule in Group 3 were aggregated to form a membrane-like layer that occludes the tubule.

**3.4.2. Occluding area ratio análisis.** Table 2 shows the proportion of the occluded dentinal tubule before and after the 1 minute exposure to 6 wt.% citric acid in each group. An independent t-test revealed that there is no statistically significant difference after the application of 6 wt.% citric acid between the dense BGN and MBGN groups (p>0.01). After the application of 6 wt.% citric acid, the proportion of the occluded dentinal tubule decreased in each group. The degree of change was smaller in the MBGN group compared to the dense BGN group, but not to a statistically significant extent (P>0.01).

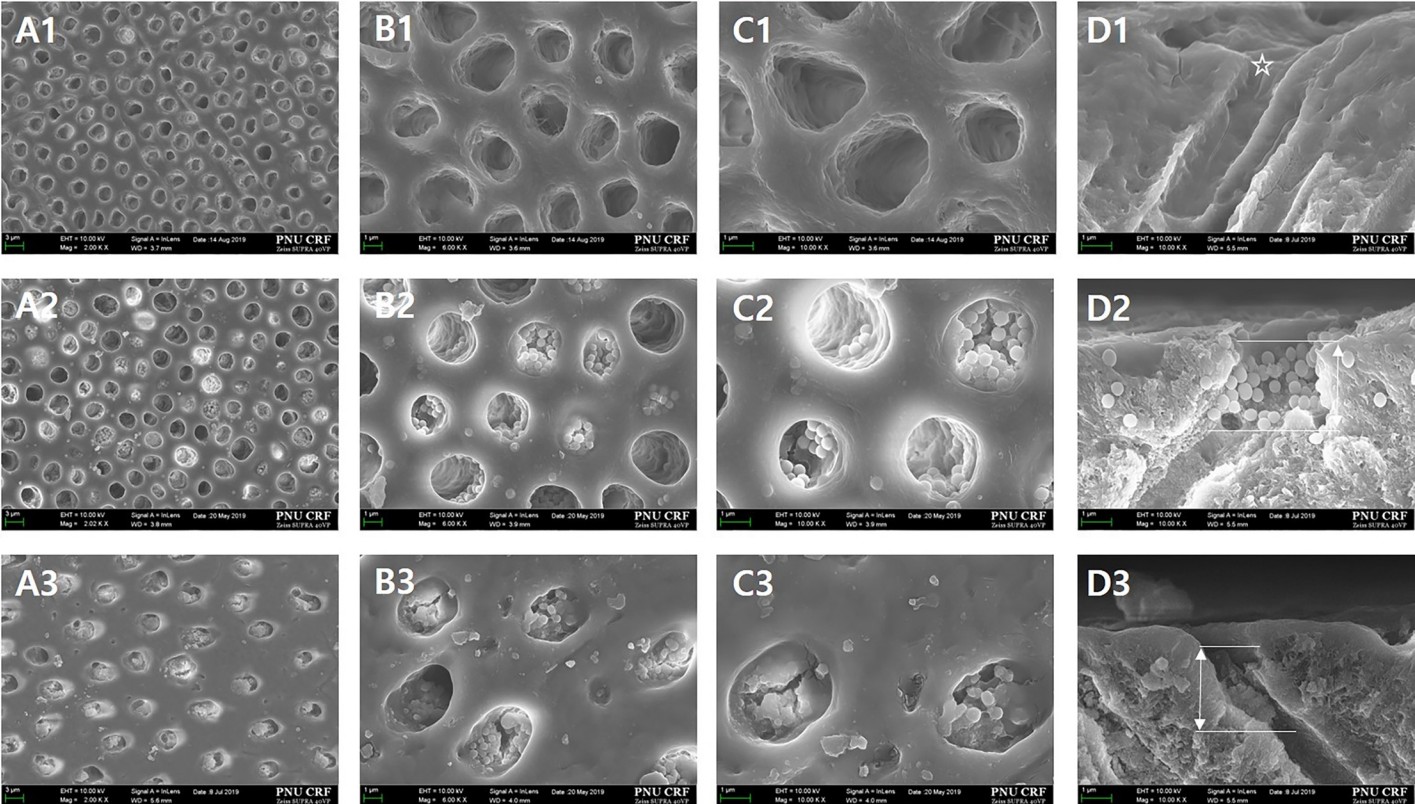

**Fig 12. FESEM micrographs of the top and FESEM micrographs to examine the tubule-occluding effects in the hypersensitive tooth model after 1 min exposure to 6 wt.% citric acid (A, x2,000; B, x6,000; C, x10,000; D, sagittal).** A1–D1 images of Group 1 (control) show that the diameter of the dentinal tubule increased (*). The image of Group 2 (dense BGN) shows that the concentration and depth of particles blocking the dentinal tubule have decreased. The image of Group 3 (MBGN) shows that there is a lower concentration of particles blocking the dentinal tubule and that the depth within the tubule has also decreased. Compared to those in Group 2, the particles in Group 3 aggregated and formed a membrane-like layer or box.

### 3.5. MTBS test

Fig 13 shows the mean and standard deviation of MTBS for each group. The Kruskal–Wallis test confirmed that there is no statistically significant difference among the control, dense BGN, and MBGN groups ($P > 0.05$).

### 3.6. Cytotoxicity assay

Table 3 present a graph of relative cell viability of HDPCs exposed to MBGN and dense BGN. Cell viability was analyzed at the same concentration of dense BGN and MBGN using a 2-sample t-test, and there was no statistically significant difference ($P > 0.05$). When the

**Table 2. Occluded dentinal tubule ratio analysis.**

| Groups | Occluded area (%) Before acid challenge | After acid challenge | P-value |
|---|---|---|---|
| Control | - | - | - |
| Dense BGN | 75.47±4.12 | 55.34±8.15 | 0.06 |
| MBGN | 67.67±7.32 | 61.43±4.03 | 0.37 |
| P-value | 0.45 | 0.29 | |

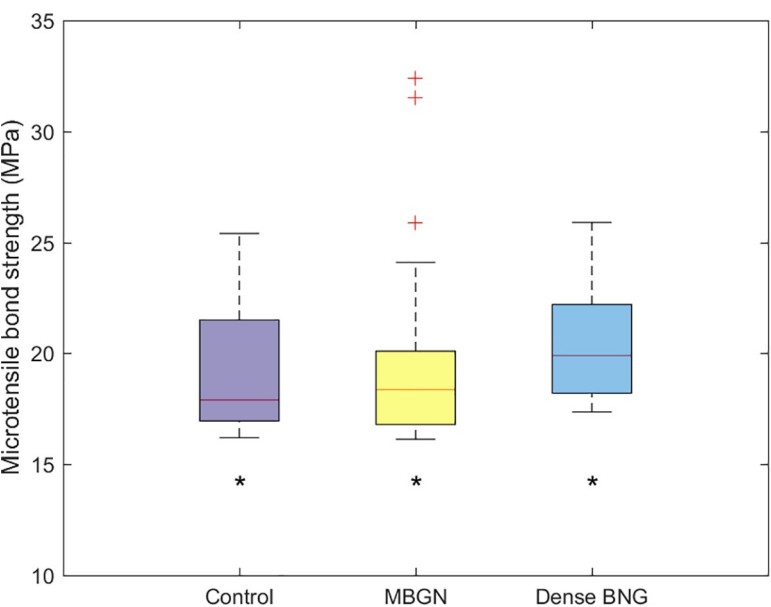

**Fig 13. Microtensile bond strength of control, dense BGN, MBGN.** Superscript letter '*' indicates no significant difference (p>0.05).

concentrations of MBGN and dense BGN were varied, the Kruskal–Wallis H test showed that there were statistically significant differences in cell viability among the groups at a dense BGN concentration of 40 μg/mL and 80 μg/mL (P<0.05). However, there were no statistically significant differences in cell viability within the groups at the remaining concentrations of dense BGN and MBGN (P>0.05).

## 4. Discussion

DH generally occurs in the presence of external stimuli in an exposed dentinal tubule. Tubule-occluding materials have been introduced as a treatment to alleviate DH, and many recent studies have investigated the use of BGN as a tubule-occluding material. Yu et al. confirmed that BGN effectively occludes an exposed dentinal tubule [16]. Porous BG is anticipated to have a greater tubule-occluding effect, although studies that directly investigated the impact of mesopores on the tubule occlusion efficacy of BG are rare. This study aims to synthesize dense BGN, which does not have a mesoporous structure, and MBGN, which does have a mesoporous structure, and compare their dentinal tubule occlusion and bioactivity.

In this study, we synthesized MBGN and dense BGN by altering the hexadecyltrimethylammonium bromide (CTAB) content based on the sol-gel synthesis protocol. XRD data of MBGN showed a peak near 2.5°, and this suggests that MBGN is a hexagonal mesoporous

**Table 3. Relative cell viability of human dental pulp cell exposed to MBGN, dens BGN at different concentration (0–320μg/mL) for 72 hours.**

| Concentration (μg/mL) Relative viability(%) | Control | 10 | 20 | 40 | 80 | 160 | 320 |
|---|---|---|---|---|---|---|---|
| Dense BGN | 100 | 104.04±0.78ab | 95.19±3.11ab | 87.16±5.48b | 107.69±2.64a | 96.82±3.88ab | 95.20±6.62ab |
| MBGN | 100 | 95.33±2.24a | 90.32±8.53 a | 92.92±4.26 a | 108.64±0.87 a | 94.14±1.90 a | 91.80±2.96 a |
| p-value | - | 0.03 | 0.41 | 0.22 | 0.59 | 0.34 | 0.46 |

Same superscript lowercase letter within rows indicates no significant difference (p>0.05).

structure. In contrast, dense BGN did not have a peak near 2.5°, which confirms the absence of mesopores within dense BGN (Fig 3). When the synthesized materials were analyzed with FTIR, silicate adsorption bands were observed, which suggests that silicate is a core material. In the $N_2$ adsorption analysis, the MBGN exhibited a type IV isotherm, suggesting that there are mesopores within the material. Dense BGN exhibited a type II isotherm suggesting that the material is non-porous, which is consistent with the XRD data. BET and BJH methods used to analyze the pore characteristics of MBGN and dense BGN showed that MBGN has a larger surface area (152.40) with a large pore size compared to dense BGN and has type IV isotherm-H1 hysteresis loops, confirming that it is a material with evenly distributed pores. These structural features of dense BGN and MBGN were also confirmed in FESEM images (Fig 6), where MBGN had an irregular surface, while dense BGN had a relatively smooth surface. The irregularity of the surface of MBGN is speculated to be due to the presence of mesopores on the surface, which contains high $Ca^{2+}$ and $PO_4^{3-}$ ion concentrations that would contribute to the material's hydroxyapatite forming ability.

In this study, we prepared tablet samples and placed them in an SBF solution to determine the level of ion dissolution and hydroxyapatite deposition on the tablet. We confirmed with XRD, FTIR, and FESEM images that the SBF solution with MBGN had a greater reduction of calcium ion concentration, and only the MBGN tablets formed hydroxyapatite (Figs 8A, 9A and 10). These results are attributable to the fact that mesopores in MBGN increased the calcium ions and phosphate ions around the particles and served as an initiation site for forming hydroxyapatite crystals [19]. In the previous studies, Si-OH (silanol) group was formed after breakage of Si-O-Si bond when silica was exposed to the SBF [24]. First silicate ions in silanol groups were combined with calcium ions in SBF to form a calcium silicate. Then calcium ions in calcium silicate combined with phosphate ions in SBF to form an amorphous calcium phosphate. This amorphous calcium phosphate was transformed into crystalline apatite after Ca/P atomic ratio was increased [25]. According to Xia et al. [26], a highly ordered mesoporous structure is important for bioactivity. Silanols (Si-OH) on the surface of BGs play an important role in hydroxyapatite deposition, and MBGNs with a large surface area have more exposed Si-OH, which leads to greater hydroxyapatite deposition [24, 27, 28]. Hydroxyapatite formation was not observed in dense BGNs after soaking in SBF solution for 42 days (Figs 8B, 9B and 10), presumably because they have a smaller surface area, fewer Si-OH exposed, and the calcium and phosphate ion concentration is inadequate.

To observe the dentinal tubule-occluding effects of MBGN and dense BGN, we prepared sensitive tooth models with exposed dentinal tubules using 1 wt.% citric acid, treated them with each material, and observed the surfaces and longitudinal sections (Fig 11). FESEM images in Fig 11 showed that both dense BGN and MBGN treatment occluded the tubule in the sensitive tooth model. In terms of the occlusion pattern, MBGN had greater aggregation of particles compared to dense BGN, but there was no statistically significant difference in the dentinal tubule occluded area between the two materials. To reflect the modern dietary style characterized by frequent consumption of acidic drinks, such as carbonated drinks and fruit juices, we treated disc tooth samples with 6 wt.% citric acid and observed the surface and longitudinal sections of the samples using FESEM (Fig 12). The FESEM image of the control group showed an increased area of the opened dentinal tubule (Fig 12 A1-D1). Both the dense BGN and MBGN groups showed a reduced occluded area and depth after 6 wt.% citric acid treatment compared to that before treatment, and the degree of reduction was greater in the dense BGN group (Fig 12A2–12D2). In MBGN-treated samples, the particles aggregated and formed a membrane-like layer, and this layer is thought to have increased the resistance to citric acid (Fig 12A3–12D3). Considering the in vitro bioactivity results, the occlusion patterns of MBGN might be related to the chemical interaction between HA on MBGN particles and dentinal

tubule surface. Such a trend is consistent with the findings of Chen et al. [29], where MBGN with hardening agent phosphoric acid led to better remineralized crystal formation owing to the greater bridging effect among particles compared to the BG group under equal conditions. Considering that the DH-relieving effect is proportional to the ability to occlude the opened dentinal tubule, the high bioactivity and bridging effect among particles of MBGN is anticipated to have strong DH-relieving effects by effectively occluding the tubule.

Many patients who developed tooth sensitivity as a result of dentinal tubule exposure undergo resin restoration treatment in the future. Therefore, it is imperative that the dense BGN and MBGN synthesized in this study do not have a negative impact on the bonding strength between dentin and adhesive (Fig 13). We bonded resin to the dense BGN- and MBGN-treated tooth samples using self-etch adhesive and measured MTBS values. The results showed that there were no statistically significant differences in MTBS values among the untreated control group and the dense BGN-treated and MBGN-treated groups (P>0.05). This contradicts the findings of Adebayo et al. [30] that the use of agents with tubule-occluding tendencies to treat DH hinders resin tag formation and diminishes bond strength. The self-etch adhesive used in this study was Clearfil SE bond, and in addition to forming mechanical interlocking by forming tags within the dentinal tubule, this product chemically binds to calcium exposed by acidic monomer 10-methacryloyloxydecyl dihydrogen phosphate (10-MDP) [31]. Because of this adhesion mechanism, applying dense BGN and MBGN to the tooth surface seems to have had no grave impact on the final MTBS values. This study only investigated the effects of dentinal tubule occlusion agents (dense BGN and MBGN) on the immediate MTBS of adhesive, and additional studies are needed to examine the effects of each agent during aging, such as that induced by thermocycling.

To use MBGN and dense BGN as DH treating agents, they must have low cell cytotoxicity because dental materials are supposed to contact directly with a tooth and its surrounding tissues [32]. In our study, we assessed cell viability via MTT assay by applying MBGN and dense BGN to human periodontal ligament (HPDL) cells. The MTT assay results confirmed that all concentrations of dense BGN and MBGN did not induce cell death of 20% or greater, so the materials are deemed biocompatible. However, more study is needed to investigate the cell cytotoxicity of individual components of both bioactive glasses.

This study found that both dense BGN and MBGN are effective dentinal tubule occlusion agents and that MBGN has dentin remineralization ability by facilitating hydroxyapatite deposition. Based on these facts, clinicians may consider the use of MBGN particles as a treatment option for prolonged DH. However, as this study was only conducted in vitro, additional in vivo studies are needed to evaluate the tubule occlusion durability, remineralization ability, and biocompatibility of MBGN.

## 5. Conclusions

In this article, we confirmed both BGN and MBGN occluded dentinal tubule mechanically. The higher bioactivity of MBGN compared with that of dense BGN arises from the structural difference, and it is anticipated to facilitate dentin remineralization by inducing hydroxyapatite deposition within the tubule and thus have higher DH-relieving effects. Therefore, BGN with mesoporous microsurface structure (MBGN) may be considered to be more effective to occlude dentinal tubule as dentin desensitizer.

## Supporting information

**S1 File. XRD raw data of dense BGN.**
(TXT)

**S2 File. XRD raw data of MBGN.**
(TXT)

**S3 File. Translation certificate.**
(PDF)

## Author Contributions

**Conceptualization:** Sung-Ae Son.

**Data curation:** Moon-Kyoung Bae, Jeong-Kil Park.

**Formal analysis:** Moon-Kyoung Bae, Yong-Il Kim, Sung-Ae Son.

**Investigation:** Yang-Jung Choi.

**Methodology:** Yang-Jung Choi, Yong-Il Kim, Sung-Ae Son.

**Project administration:** Sung-Ae Son.

**Supervision:** Yong-Il Kim, Sung-Ae Son.

**Visualization:** Sung-Ae Son.

**Writing – original draft:** Yang-Jung Choi, Sung-Ae Son.

**Writing – review & editing:** Jeong-Kil Park, Sung-Ae Son.

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
