## [Decision Letter · Decision Letter 0]

11 Jun 2020

PONE-D-20-07389

Effects of microsurface structure of bioactive nanoparticles on dentinal tubules as a dentin desensitizer

PLOS ONE

Dear Dr. Son,

Thank you for submitting your manuscript to PLOS ONE. After careful consideration, we feel that it has merit but does not fully meet PLOS ONE’s publication criteria as it currently stands. Therefore, we invite you to submit a revised version of the manuscript that addresses the points raised during the review process.

It has been reviewed by experts in the field and we request that you make Moderate Revisions before it is processed further. Please find your manuscript and the review reports at the following link:

Reviewer & Editor Comments

This contribution would require moderate revision before it could be reconsidered for publication in PONE. Some of my comments and questions on this manuscript are as follows:

Q1: Section including “Introduction”; “Materials and methods”; “Results”; “Discussion” and corresponding subsection is required numbering.

Q2: Some of subsections should be merged together.

Q3: “FESEM evaluation of dentinal tubule occlusion” section should be shifted to the “Materials characterizations” section.

Q4: The authors should have a consistent font size throughout the manuscript; surprisingly some part of the main text is in a BOLD type.

Q5: The author stated that “MBGNs with a large surface area have more exposed Si-OH, which leads to greater hydroxyapatite deposition [25, 26, 27]”. In this regard, the author should further highlighted the role of Si and OH ions on the acceleration of apatite deposition when MBGNs exposed to the physiological solution

Q6: I strongly recommended the author present the a schematic image regarding the preparation, and characterization of bioactive glass nanoparticle prepared by sol-gel method (Fig. 1).

Q7: The quality of the FTIR spectra of MBGN and dense BGN (Fig. 3) should be improved.

Q8: Fig. 6, Fig. 7 and Fig. 8 should be merged together and presented as Fig. 6a, b and c. The Y and X axis of all figures beside their quality should be significantly improved. ALL figures should be replotted.

Q9: The quality of Fig. 14 should be improved.

Q10: Surprisingly small reference to PLOS ONE in the literature despite the large relevant literature there. This should be improved. There are several important papers in the recent literature of PLOS ONE.

“[6,7,8,9]” should replace with “[6-9]”

“[25, 26, 27]” should replace with “[25-27]”

We look forward to receiving your revised manuscript.

Kind regards,

Hamid Reza Bakhsheshi-Rad

Academic Editor

PLOS ONE

Journal Requirements:

2. Thank you for providing information about the ethics approval and informed patient consent.

However, we additionally ask that you update your Methods section to specify what type of consent was given (i.e., written, verbal, etc.)

We also ask you to provide the number of the IRB-approved protocol in both the Methods and the Ethics statement, if available.

3. Please ensure that you refer to Figure 1 in your text as, if accepted, production will need this reference to link the reader to the figure.

Reviewers' comments:

Reviewer's Responses to Questions

**Comments to the Author**

1. Is the manuscript technically sound, and do the data support the conclusions?

Reviewer #1: Yes

2. Has the statistical analysis been performed appropriately and rigorously? 

Reviewer #1: Yes

3. Have the authors made all data underlying the findings in their manuscript fully available?

Reviewer #1: Yes

4. Is the manuscript presented in an intelligible fashion and written in standard English?

Reviewer #1: No

5. Review Comments to the Author

Reviewer #1: The manuscript entitled “Effects of microsurface structure of bioactive nanoparticles on dentinal tubules as a dentin desensitizer” by Yang-Jung Choi et al is a nice submission on DH. The manuscript is acceptable with some minor changes

1. Combine line 2 and 3.

2. Abstract is too long and contain unnecessary information.

3. Experimental detail about the individual component cytotoxicity should be given.

4. Figure 1 need to be reconstructed in a visual experimental form rather than like a flow chart.

5. Figure 2A, Provide detail of peak in the picture.

6. Figure 6,7, 8 need to be made more stylish to look good for the readers.

7. Other than these minor flaws, author should also correct the English mistakes which are often present in the manuscript.

8. Recent literatures need to be cited. Some suggested one are: DOI: 10.1021/acs.chemrestox.8b00129, DOI: 10.1039/c8tx00112j

6. PLOS authors have the option to publish the peer review history of their article (what does this mean?). If published, this will include your full peer review and any attached files.

Reviewer #1: No

---

## [Author Response · Author response to Decision Letter 0]

19 Jul 2020

Reviewer & Editor Comments

This contribution would require moderate revision before it could be reconsidered for publication in PONE. Some of my comments and questions on this manuscript are as follows:

Q1: Section including “Introduction”; “Materials and methods”; “Results”; “Discussion” and corresponding subsection is required numbering.

A1(Answer1): We added numbering to each section and corresponding subsection.

Q2: Some of subsections should be merged together.

A2: We have merged some subsections. Please check the revised file.

In “materials and methods” section, “Synthesis of dense bioactive glass nanoparticles” and “Synthesis of mesoporous bioactive glass nanoparticles” were combined to “Synthesis of dense bioactive glass nanoparticles and mesoporous bioactive glass nanoparticles”. In “Results” section, “XRD” and “FTIR” were combined to “XRD and FTIR”. 

Q3: “FESEM evaluation of dentinal tubule occlusion” section should be shifted to the “Materials characterizations” section.

A3: We thought that applying the bioactive glass on the dentinal tubules to evaluate the dentinal tubules' sealing capacity was part of the clinical application of bioactive materials. Therefore, instead of shifting to the 'materials characterization' section, it would be better to place it as a separate subtitle to highlight its clinical relevance.

Q4: The authors should have a consistent font size throughout the manuscript; surprisingly some part of the main text is in a BOLD type.

A4: We checked the whole manuscript thoroughly and corrected the wrong text type. 

Q5: The author stated that “MBGNs with a large surface area have more exposed Si-OH, which leads to greater hydroxyapatite deposition [25, 26, 27]”. In this regard, the author should further highlighted the role of Si and OH ions on the acceleration of apatite deposition when MBGNs exposed to the physiological solution

A5: Thanks for pointing out. We added additional information regarding the role of Si and OH ions on the acceleration of apatite deposition when MBGN is exposed to physiological solutions. The added contents are as follows. : 1)The first silicate ion of the Si-OH group was combined with the calcium ion in the SBF solution. 2)In addition, calcium ions of calcium silicate combine with phosphate ions of SBF to form amorphous calcium phosphate. 3)This amorphous calcium phosphate was transformed into crystalline apatite. 4)MBGN with a large surface area had more exposed Si-OH and this feature accelerated the deposition of hydroxyapatite. Check the revised manuscript.

Q6: I strongly recommended the author present the a schematic image regarding the preparation, and characterization of bioactive glass nanoparticle prepared by sol-gel method (Fig. 1).

A6: We added schematic images of the fabrication and characterization of bioactive glass nanoparticles and schematic images of material methods (Fig. 1).

Q7: The quality of the FTIR spectra of MBGN and dense BGN (Fig. 3) should be improved.

A7: We replotted most figures, including FTIR spectra of MBGN and dense BGN. (Fig. 4)

Q8: Fig. 6, Fig. 7 and Fig. 8 should be merged together and presented as Fig. 6a, b and c. The Y and X axis of all figures beside their quality should be significantly improved. ALL figures should be replotted.

A8: We agree that Figures 6, 7, and 8 should be combined. We combined Figures 6, 7, and 8 into Figure 6.

Q9: The quality of Fig. 14 should be improved.

A9: We replotted Fig. 14 to improve the quality of the graph.

Q10: Surprisingly small reference to PLOS ONE in the literature despite the large relevant literature there. This should be improved. There are several important papers in the recent literature of PLOS ONE.

“[6,7,8,9]” should replace with “[6-9]”

“[25, 26, 27]” should replace with “[25-27]”

A10: We could find the recent literature of PLOS ONE about bioactive glasses and added two more references published in PLOS ONE(reference 9,21). Also, we have reformatted the reference part in the manuscript according to the above style guidelines.

Journal Requirements:

A1: We have reformatted the manuscript according to the above style guidelines.

2. Thank you for providing information about the ethics approval and informed patient consent.

However, we additionally ask that you update your Methods section to specify what type of consent was given (i.e., written, verbal, etc.)

We also ask you to provide the number of the IRB-approved protocol in both the Methods and the Ethics statement, if available.

A2: Thank you for your concern about ethical approval and your consent for the study. Our study was a deliberation exemption study. The deliberation exemption criteria included: Researchers do not know the personally identifiable information of human-derived donors, and the results obtained through the study are not related to the donor's genetic characteristics. Therefore, we conducted a study based on donor consent exemption. We revised the information regarding the plaintiff's ethical approval and prepared and posted an ethical statement file.

3. Please ensure that you refer to Figure 1 in your text as, if accepted, production will need this reference to link the reader to the figure.

A3: We agree to offer link of the figure to the reader. Let us know if we need to revise more.

Review Comments to the Author

Reviewer #1: The manuscript entitled “Effects of microsurface structure of bioactive nanoparticles on dentinal tubules as a dentin desensitizer” by Yang-Jung Choi et al is a nice submission on DH. The manuscript is acceptable with some minor changes

1. Combine line 2 and 3.

We have combined the line 2 and 3 in the first page. 

2. Abstract is too long and contain unnecessary information.

We agree that the Abstract is too long and contains unnecessary information. We have removed unnecessary experimental details and shortened the length of the Abstract. 

3. Experimental detail about the individual component cytotoxicity should be given.

Thank you for the reviewer's request for experimental details of individual component cytotoxicity. In our experiments, we investigated cell viability after direct exposure to bioactive glasses according to ISO 10993-5. We have not investigated the cytotoxicity of individual components of bioactive glasses, which may be a limitation of the study. To clarify the individual component cytotoxicity of bioactive glasses, we believe more research is needed. We have added comments on the limitations of our manuscript.

4. Figure 1 need to be reconstructed in a visual experimental form rather than like a flow chart.

We have changed the type of figure from the flow chart to an experimental visual form. Please check the figure 2. 

5. Figure 2A, Provide detail of peak in the picture.

We acknowledge that usually, XRD graph includes detail of peak to specify the crystal structure of materials. Since our specimen (bioactive glasses) had an amorphous structure, we concluded that it was not meant to provide detail of peak in the picture. We attached the raw data of the materials (S1, S2 file). Please check. 

6. Figure 6,7, 8 need to be made more stylish to look good for the readers.

We agree that the Fig. 6, 7, 8 need to be made more stylish. Therefore, we replotted most figures. We combined Fig. 6, 7, 8 express the result of ion dissolution in the one figure (Fig 6.)

7. Other than these minor flaws, author should also correct the English mistakes which are often present in the manuscript.

We have checked the manuscript again. Also, we added a translation certificate(S3 file).

8. Recent literatures need to be cited. Some suggested one are: DOI: 10.1021/acs.chemrestox.8b00129, DOI: 10.1039/c8tx00112j

 We have added a new reference at the discussion part.

---

## [Editor Report · Decision Letter 1]

3 Aug 2020

Effects of microsurface structure of bioactive nanoparticles on dentinal tubules as a dentin desensitizer

PONE-D-20-07389R1

Dear Dr. Son,

We’re pleased to inform you that your manuscript has been judged scientifically suitable for publication and will be formally accepted for publication once it meets all outstanding technical requirements.

Kind regards,

Hamid Reza Bakhsheshi-Rad

Academic Editor

PLOS ONE
---

## [Editor Report · Acceptance letter]

6 Aug 2020

PONE-D-20-07389R1 

Effects of microsurface structure of bioactive nanoparticles on dentinal tubules as a dentin desensitizer 

Dear Dr. Son:

I'm pleased to inform you that your manuscript has been deemed suitable for publication in PLOS ONE. Congratulations! Your manuscript is now with our production department. 

Kind regards, 

on behalf of

Dr. Hamid Reza Bakhsheshi-Rad 

Academic Editor

PLOS ONE